# How geographic access to care shapes disease burden: The current impact of post-exposure prophylaxis and potential for expanded access to prevent human rabies deaths in Madagascar

**Malavika Rajeev**[1]*, **Hélène Guis**[2,3,4,5], **Glenn Torrencelli Edosoa**[6], **Chantal Hanitriniaina**[7], **Anjasoa Randrianarijaona**[3], **Reziky Tiandraza Mangahasimbola**[3], **Fleur Hierink**[8,9], **Ravo Ramiandrasoa**[10], **José Nely**[11], **Jean-Michel Heraud**[12], **Soa Fy Andriamandimby**[12], **Laurence Baril**[3¤‡], **C. Jessica E. Metcalf**[1‡], **Katie Hampson**[13‡]

1 Department of Ecology and Evolutionary Biology, Princeton University, Princeton, New Jersey, United States, 2 CIRAD, UMR ASTRE, Antananarivo, Madagascar, 3 Epidemiology and Clinical Research Unit, Institut Pasteur de Madagascar, Antananarivo, Madagascar, 4 ASTRE, Univ Montpellier, CIRAD, INRAE, Montpellier, France, 5 FOFIFA-DRZVP, Antananarivo, Madagascar, 6 Chargé des Maladies Tropicales Négligées, Organisation mondiale de la Santé Madagascar, Antananarivo, Madagascar, 7 Mention Zoologie et Biodiversité Animale, Faculté des Sciences, Université d'Antananarivo, Antananarivo, Madagascar, 8 Institute of Global Health, Faculty of Medicine, University of Geneva, Geneva, Switzerland, 9 Institute for Environmental Sciences, University of Geneva, Geneva, Switzerland, 10 Service Contre les Maladies Endémo-épidémiques et Tropicales Négligées, Ministère de la Santé Publique, Antananarivo, Madagascar, 11 Vaccination Center, Institut Pasteur de Madagascar, Antananarivo, Madagascar, 12 Virology Unit, Institut Pasteur de Madagascar, Antananarivo, Madagascar, 13 Boyd Orr Centre for Population and Ecosystem Health, Institute of Biodiversity, Animal Health and Comparative Medicine, University of Glasgow, Glasgow, United Kingdom

¤ Current address: Institut Pasteur du Cambodge, Phnom Penh, Cambodia
‡ These authors are joint senior authors on this work.
* mrajeev@princeton.edu

## Abstract

### Background

Post-exposure prophylaxis (PEP) is highly effective at preventing human rabies deaths, however access to PEP is limited in many rabies endemic countries. The 2018 decision by Gavi to add human rabies vaccine to its investment portfolio should expand PEP availability and reduce rabies deaths. We explore how geographic access to PEP impacts the rabies burden in Madagascar and the potential benefits of improved provisioning.

### Methodology & principal findings

We use spatially resolved data on numbers of bite patients seeking PEP across Madagascar and estimates of travel times to the closest clinic providing PEP (N = 31) in a Bayesian regression framework to estimate how geographic access predicts reported bite incidence. We find that travel times strongly predict reported bite incidence across the country. Using resulting estimates in an adapted decision tree, we extrapolate rabies deaths and reporting

**Data Availability Statement:** All processed data, code, and outputs are archived on Zenodo (http://doi.org/10.5281/zenodo.4064312 and https://doi.org/10.5281/zenodo.4064304). The raw bite patient data at the national level are maintained in two secure REDCap (project-redcap.org) databases, one for IPM and another for all peripheral clinics provisioning PEP. These databases were last queried on September 19, 2019 for these analyses. The IPM GIS unit provided the data on geolocated clinics across the country. Anonymized raw bite patient data and full data on geolocated clinics are available from IPM following institutional data transfer protocols (contact: rrandrem@pasteur.mg) Anonymized raw data collected from the Moramanga District were retrieved from the Wise Monkey Portal (wisemonkeyfoundation.org) on the same date and are shared in the archived repository.

**Funding:** This work was funded by grants from the Center for Health and Wellbeing and the Department of Ecology and Evolutionary Biology at Princeton University to CJEM and MR. MR is supported by an NSF Graduate Research Fellowship. KH is supported by the Wellcome Trust (207569/Z/17/Z). Funding for open access publication was made possible by support from the Princeton University Library Open Access Fund. The funders had no role in study design, data collection and analysis, decision to publish, or preparation of the manuscript.

**Competing interests:** The authors have declared that no competing interests exist.

and find that geographic access to PEP shapes burden sub-nationally. We estimate 960 human rabies deaths annually (95% Prediction Intervals (PI): 790–1120), with PEP averting an additional 800 deaths (95% PI: 640–970) each year. Under these assumptions, we find that expanding PEP to one clinic per district (83 additional clinics) could reduce deaths by 19%, but even with all major primary clinics provisioning PEP (1733 additional clinics), we still expect substantial rabies mortality. Our quantitative estimates are most sensitive to assumptions of underlying rabies exposure incidence, but qualitative patterns of the impacts of travel times and expanded PEP access are robust.

## Conclusions & significance

PEP is effective at preventing rabies deaths, and in the absence of strong surveillance, targeting underserved populations may be the most equitable way to provision PEP. Given the potential for countries to use Gavi funding to expand access to PEP in the coming years, this framework could be used as a first step to guide expansion and improve targeting of interventions in similar endemic settings where PEP access is geographically restricted and baseline data on rabies risk is lacking. While better PEP access should save many lives, improved outreach, surveillance, and dog vaccination will be necessary, and if rolled out with Gavi investment, could catalyze progress towards achieving zero rabies deaths.

## Author summary

Canine rabies causes an estimated 60,000 deaths each year across the world, primarily in low- and middle-income countries where people have limited access to both human vaccines (post-exposure prophylaxis or PEP) and dog rabies vaccines. Given that we have the tools to prevent rabies deaths, a global target has been set to eliminate deaths due to canine rabies by 2030, and recently, Gavi, a multilateral organization that aims to improve access to vaccines in the poorest countries, added human rabies vaccine to it's portfolio. In this study, we estimated reported incidence of patients seeking PEP in relation to travel times to clinics provisioning PEP and extrapolate human rabies deaths in Madagascar. We find that PEP currently averts around 800 deaths each year, but that the burden remains high (1000 deaths/year), particularly in remote, hard-to-reach areas. We show that expanding PEP availability to more clinics could significantly reduce rabies deaths in Madagascar, but our results reaffirm that expansion alone is will not achieve the global goal of zero human deaths from dog-mediated rabies by 2030. Combining PEP expansion with outreach, surveillance, and mass dog vaccination programs will be necessary to move Madagascar, and other Low- and Middle-Income countries, forward on the path to rabies elimination.

## Introduction

Inequities in access to care are a major driver of disease burden globally [1]. Often, the populations at greatest risk of a given disease are the most underserved [2]. Delivering interventions to these groups is challenging due to financial and infrastructural limitations and requires careful consideration of how best to allocate limited resources [3].

Canine rabies is estimated to cause approximately 60,000 human deaths annually [4]. Mass vaccination of domestic dogs has been demonstrated to be a highly effective way to control the disease in both animals and humans. While dog vaccination can interrupt transmission in the reservoir, human deaths can also be prevented through prompt administration of post-exposure prophylactic vaccines (PEP) following a bite by a rabid animal [5]. However, access to the human rabies vaccine is limited in many countries where canine rabies is endemic [6–8], and within countries these deaths are often concentrated in rural, underserved communities [9].

In 2015, a global framework to eliminate deaths due to canine rabies by 2030 ('Zero by 30') through a combination of PEP provisioning and dog vaccination was established by the World Health Organization (WHO) and partners [10]. Furthermore, in 2018, Gavi, the Vaccine Alliance, added human rabies vaccines to their proposed investment portfolio [11]. From 2021, Gavi-eligible countries should be able to apply for support to expand access to these vaccines, with potential to greatly reduce deaths due to rabies.

A primary challenge in expanding access effectively is the lack of data on rabies exposures and deaths in humans and incidence in animals in most rabies-endemic countries [12]. Deaths due to rabies are often severely underreported, with many people dying outside of the health system, often in remote and marginalized communities [13]. Instead of directly measuring rabies deaths, the majority of rabies burden studies use bite patient data on reported bites at clinics provisioning PEP and a decision tree framework to extrapolate deaths, assuming that overall reported bite incidence (i.e. both bites due to non-rabid and rabid animals) is proportional to rabies incidence (i.e. the more bites reported in a location, the higher the incidence of rabies exposures), and that reporting to clinics for PEP is uniform across space [8,14,15]. If applied sub-nationally, these assumptions would likely underestimate rabies deaths in places with poor access to PEP and may overestimate rabies deaths in places with better access to PEP.

In Madagascar, the Institut Pasteur de Madagascar (IPM) provides PEP to 30 Ministry of Health clinics, in addition to its own vaccine clinic, where PEP is available at no direct cost to patients [15]. Other than at these 31 clinics, PEP is not available at any other public clinics or through the private sector. In addition, there is limited control of rabies in dog populations and the disease is endemic throughout the country [16,17]. Due to the spatially restricted nature of PEP provisioning and lack of direct costs for PEP, geographic access is likely to be a major driver of disease burden within the country. Previously, we estimated the burden of rabies in Madagascar using data from a single district to extrapolate to the national scale, but did not account for spatial variation in access [15]. Here, we provide revised estimates of human rabies deaths by incorporating the impact of access to PEP at the sub-national level on preventing human rabies deaths and explore the potential impact of expanding provisioning of human rabies vaccines on further reducing these deaths. This framework may usefully apply to other countries where PEP availability is currently geographically restricted in considering how to most effectively and equitably provision these life-saving vaccines.

## Methods

### Ethics statement

Data collection from the Moramanga District was approved by the Princeton University IRB (7801) and the Madagascar Ministry of Public Health Ethics Committee (105-MSANP/CE). Oral informed consent was obtained from all interviewed participants. Data collected from bite patients at the national level are maintained jointly by the Ministry of Health and IPM as a routine part of PEP provisioning.

### Estimating geographic access to PEP

To estimate mean and population weighted travel times to the nearest clinic, we used two raster datasets: 1) the friction surface from the Malaria Atlas Project [18] at an ~1 km$^2$ scale and 2) the population estimates from the 2015 UN adjusted population projections from World Pop ([19], originally at an ~100m$^2$ resolution), which we aggregated to the friction surface (Fig A in S1 Text).

From GPS locations of the 31 clinics that currently provision PEP, we estimated the travel time to the nearest clinic at an approximately 1 x 1 km scale as described in [18]. We then extracted the mean and population-weighted mean travel times for each district (2nd level administrative division, N = 114) and commune (the administrative unit below the district, N = 1579), and Euclidean distance, i.e. the minimum distance from the administrative unit centroid to any clinic. We used shapefiles from the UN Office for the Coordination of Humanitarian Affairs (OCHA) available through the Humanitarian Data Exchange for the district and communes boundaries (retrieved on October 31, 2018, https://data.humdata.org/dataset/madagascar-administrative-level-0-4-boundaries). To see which metric best predicted ground-truthed travel time data, we compared travel times and distance estimates to driving times collected by IPM during field missions, i.e. time it took to travel by car between two locations excluding break times (N = 43), and patient reported travel times from a subset of Moramanga clinic bite patients (N = 1057), see Fig B in S1 Text for raw data) by seeing which had the highest R$^2$ when predicting estimated travel times in a linear model.

### Estimating bite incidence

We used two datasets on bite patients reporting to clinics for PEP:

A national database of individual bite patient forms from the 31 clinics provisioning PEP across the country between 2014–2017. These forms were submitted to IPM with frequencies ranging from monthly to annually, and included the patient reporting date as well as the patient residence (i.e. the district where the patient lived).

33 months of data (between October 2016 and June 2019) on patients reporting to the Moramanga clinic resolved to the commune level.

For the national data, some clinics did not submit any data, or had substantial periods (months to a whole year) with no submitted data. To correct for this, we exclude periods of 15 consecutive days with zero submitted records (see S2 Text). For each clinic we divided the total number of bites reported in a given year by the estimated proportion of forms which were not submitted (i.e., under-submission). Due to yearly variation in submissions, we took the average of annual bite incidence estimates aggregated to district level. We validated this approach by comparing estimated vial demand given the total reported bites corrected for under-submission to vials provisioned to clinics for 2014–2017 (see S2 Text). At both the commune and district administrative level, we assigned clinic catchments by determining which were closest in terms of travel times for the majority of the population within the administrative unit. For national data, we excluded any districts in a catchment of a clinic which submitted less than 10 forms and any years for which we estimated less than 25% of forms were submitted.

### Modeling reported bite incidence as a function of access

We modeled the number of reported bites as a function of travel time (*T*) using a Poisson regression:

$$\mu_i = e^{(\beta_t T_i + \beta_0)} P_i$$

$$y_i = Poisson(\mu_i)$$

where $y_i$ is the average number of bites reported to a clinic annually and $\mu_i$ the expected number of bite patients presenting at the clinic as a function of travel time ($T_i$) and human population size ($P_i$) (an offset which scales the incidence to the expected number of bites) for a given source location $i$ (district or commune). We fit this model to both the national data (district level) and the Moramanga data (commune level). To more directly compare estimates between datasets, we also modeled the national data with a latent commune-level travel time covariate ($T_j$):

$$\mu_i = \sum_{j=1}^{j} e^{(\beta_t T_j + \beta_{0j})} P_j$$

As travel times are correlated with population size (Fig A in S3 Text), we also compared how well bites were predicted by population size alone, and in combination with travel times. For the models with population size, we removed the offset and used either population size alone ($\mu_i = e^{(\beta_p P_i + \beta_0)}$) or population size and travel times ($\mu_i = e^{(\beta_t T_i + \beta_p P_i + \beta_0)}$) as predictors.

For the models fit to the national data, we also modeled variation between clinics with a catchment random effect: $B_{0,k} \sim norm(\mu, \sigma_0)$), where $\mu$ is the mean and $\sigma_0$ is standard deviation and $B_{0,k}$ is the catchment level intercept.

We tested whether the catchment random effect captured overdispersion in the data (i.e. whether variance equals the mean, the expectation given a Poisson distribution) rather than any catchment specific effects by extending these models with an overdispersion parameter: $\epsilon_i \sim norm(0, \sigma_e)$, where $\sigma_e$ is the standard deviation around a random variable with mean of zero [20]:

$$\mu_i = e^{\left(\sum_{j=1}^{j} \beta_j X_j + \epsilon_i\right)} P_i$$

And where $\sum_{j=1}^{j} \beta_j X_j$ is the sum of all parameters for a given model. We fit all models in a Bayesian regression framework via MCMC using the R package 'rjags' [21]. We used model estimates to generate fitted and out-of-fit predictions and examined the sensitivity of estimates to adjustments for under-submission of forms (S3 Text).

## Modeling human rabies deaths

We estimate rabies deaths as a function of the number of bites predicted by our model and estimates of endemic rabies exposure incidence using an adapted decision tree framework. Table 1 lists all parameter values and their sources. Fig 1 describes how these parameters are used in the decision tree and the key outputs ($A_i$, deaths averted by PEP, and $D_i$, deaths due to rabies).

For $E_i$, we center the distribution at the lower end of our estimated exposure incidence from the Moramanga District (42 exposures/100,000 persons), with a range applied assuming 1% rabies incidence in dogs (estimated across a range of human-to-dog ratios between 5–25)

**Table 1. Parameters used in the decision tree to estimate human rabies deaths at the administrative level.**

| Parameter | Value | Description | Source |
|---|---|---|---|
| $B_i$ | Function of travel time to closest clinic provisioning PEP | Modeled estimates of reported bite incidence | Bayesian regression model (see Methods) |
| $E_i$ | Triangular(a = 15, b = 76, c = 42) | Annual exposures per 100,000 persons | [4,15], see Fig A in S4 Text |
| $p_{rabid}$ | Triangular(a = 0.2, b = 0.6, c = 0.4) | Proportion of reported bites that are rabies exposures[1] | [15] |
| $\rho_{max}$ | 0.98 | The maximum reporting possible for any location; data from the commune closest to the Moramanga clinic (average travel time estimate of ~ 3.12 minutes) | [15] |
| $p_{death}$ | 0.16 | The probability of death given a rabies exposure | [22] |

1 $p_{rabid}$ is constrained so that rabid reported bites cannot exceed the total expected number of rabies exposures ($E_i$) or maximum reporting in a given simulation ($\rho_{max}$).

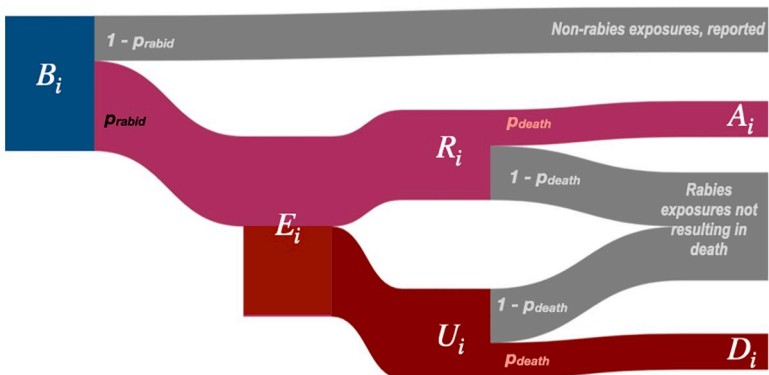

**Fig 1. Decision tree for burden estimation.** For a given administrative unit i, human deaths due to rabies ($D_i$) are calculated from model predicted reported bites ($B_i$). To get $R_i$, the number of reported bites that were rabies exposures, we multiply $B_i$ by $p_{rabid}$, the proportion of reported bites that are rabies exposures. $R_i$ is subtracted from $E_i$ to get the number of unreported bites ($U_i$) and then multiplied by the probability of death given a rabies exposure ($p_{death}$) to get deaths due to rabies ($D_i$). Similarly, deaths averted by PEP, $A_i$, are estimated by multiplying $R_i$ by $p_{death}$, i.e. those who would have died given exposure, but instead received PEP. Both $E_i$ and $p_{rabid}$ are drawn from a triangular distribution. Parameter values and sources are listed in Table 1.

and that on average a rabid dog exposes 0.39 persons [4] (see Fig A in S4 Text). As there is little data on dog population size and human exposure incidence in Madagascar [16,23], the range we used encompasses both observed human-to-dog ratios across Africa [14,24] and recent subnational estimates from Madagascar [25], and generates similar exposure incidences as observed previously across Africa [26,27]. Given previously high observed compliance in Madagascar [15], we assume that all rabies exposed patients who report to a clinic receive and complete PEP, and PEP is completely effective at preventing rabies.

## Estimating the impact of expanding PEP provisioning

We developed a framework to rank clinics by how much their PEP provision improves access for underserved communities, estimating incremental reductions in burden and increases in vaccine demand. Specifically, we aggregated our model-predicted estimates of annual bites to the clinic level. As multiple clinics may serve a single district or commune, we allocated bites to clinics according to the proportion of the population in each administrative unit which were closest to that clinic. For each clinic, we simulate throughput by randomly assigning patient presentation dates, and then assume perfect compliance (i.e. patients report for all doses) to generate subsequent vaccination dates. We use these dates to estimate vial usage given routine vial sharing practices in Madagascar [15], but assuming adoption of the WHO-recommended abridged intradermal regimen (2 x 0.1 ml injections on days 0, 3, and 7 [28]). For both burden and vial estimates, we take the mean of 1000 simulations as each clinic is added.

We simulate expansion first to each district (N = 114) and then to each commune in the country for all communes with a clinic. We select the primary clinic (primary health facility, usually with capacity to provision vaccines) in the highest density grid cell of the administrative unit as candidates for expansion. For the 85 communes without a primary clinic, we chose the secondary clinic (secondary health facility, often without formal vaccination capacity) in the highest density grid cell. 94 communes lacked any health facilities. Finally, we explore a scenario where all additional primary clinics (totaling 1733) provision PEP.

We tested three metrics for ranking additional clinics: 1) The proportion of people living >3 hours from an existing clinic that provisions PEP for which travel times were reduced; 2)

This proportion weighted by the magnitude of the change in travel times and 3) The mean reduction in travel times for people living >3 hours from an existing PEP clinic. We simulated expansion of clinics provisioning PEP to each district using these three metrics and chose the metric which decreased burden the most compared to simulations (N = 10) where clinics were added randomly to districts. For the full simulation of expanded access, once clinics reduced travel times for less than 0.01% of the population (< 2400 living greater than $x$ hrs away, starting with $x = 3$ hrs), we reduced the travel time threshold by 25%.

### Sensitivity analysis

To test the effect of our model assumptions on estimates of rabies burden and vial demand, we did a univariate sensitivity analysis of both parameters from the models of bite incidence and the decision tree (see Fig A in S6 Text for parameter ranges used). We also examined how systematic variation in rabies incidence with human population size affected burden estimates. Specifically, if human-to-dog ratios are positively correlated with human populations (i.e. dog ownership/populations are higher in more populated, urban areas), we might expect higher rabies exposure incidence as population size increases. Alternatively, if human-to-dog ratios inversely correlate with population size (i.e. dog ownership is more common in less populated, rural areas), we might expect exposure incidence to scale negatively with population size. We use scaling factors to scale incidence either positively or negatively with observed population sizes at the district and commune levels, while constraining them to the range of exposure incidence used in the main analyses (15.6–76 exposures per 100,000 persons, Fig B in S4 Text), and then simulate baseline burden, as well as expanded PEP access.

### Data and analyses

All analyses were done in R version 4.0.2 (2020-06-22) [29]] and using a number of additional packages (see S7 Text for details).

## Results

### Estimates of travel times to clinics are high and variable across Madagascar

Based on the estimates from the friction surface, approximately 36% of the population of Madagascar are estimated to live over 3 hours from a clinic (Fig 2). However, we found that these estimates underestimated both driving times across the country and patient-reported travel times to the Moramanga PEP clinic (Fig 2C). Patient reported travel times were highly variable for a given commune compared to the estimated travel times (Fig B in S1 Text). This is likely due to the fact that the friction surface assumes that the fastest available mode of transport is used across each grid cell (i.e. the presence of a road indicates that all travel through that grid cell is by vehicle), while patients reported using multiple modes of transport, with some individuals walking days to the Moramanga PEP clinic (Fig C in S1 Text).

While the travel time estimates may not reflect exact distributions of travel times, they were correlated with ground-truthed driving and patient-reported times and likely reflect patterns of access over the country (Fig 3C, and Fig D in S1 Text). Travel times weighted by population at the grid cell level were a better predictor than unweighted travel times or distance ($R^2$ = 0.43, Table A in S1 Text), therefore, we use population-weighted travel time as a proxy for access at the commune/district level in subsequent analyses.

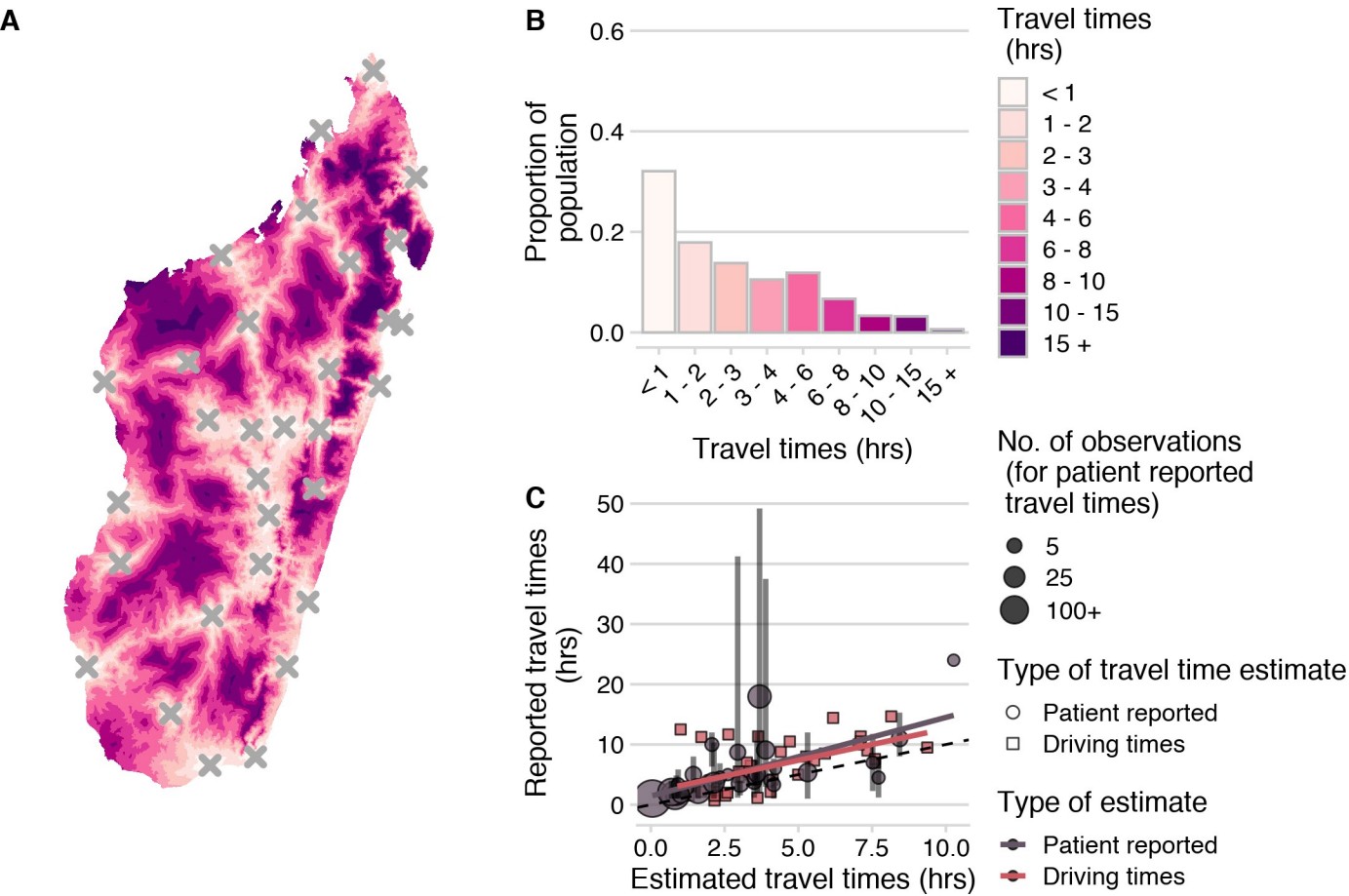

**Fig 2. Travel times to clinics provisioning PEP across Madagascar.** (A)) Estimated at an ~ 1 km2 scale using the estimated from the Malaria Atlas Project friction surface (https://malariaatlas.org/research-project/accessibility-to-cities/, CC-BY 3.0) and on human population from WorldPop (https://www.worldpop.org/geodata/summary?id=70, CC-BY 4.0). (A)) Estimated at an ~ 1 km2 scale using the estimated from the Malaria Atlas Project friction surface (https://malariaatlas.org/research-project/accessibility-to-cities/, CC-BY 3.0) and on human population from WorldPop (https://www.worldpop.org/geodata/summary?id=70, CC-BY 4.0). (B) Distribution of the population across travel times. (C) Correlation between ground-truthed travel times (mean of patient reported travel times to the Moramanga PEP clinic at the commune level and reported driving times between GPS points) and friction surface travel time estimates. The vertical lines show the 95% quantiles for reported travel times and the point size shows the number of observations for each commune. The best fit lines (red and grey) from a linear model where observed travel times are predicted by estimated travel times for each data source are also shown. The dashed black line is the 1:1 line.

### As travel times increase, reported bite incidence decreases

Bite incidence estimates generally increased with decreasing weighted travel times at both administrative scales (district and commune), although there was considerable variation between catchments for the magnitude of this relationship (Fig 3C and 3D). After additionally excluding any year with less than 25% of forms submitted, our final dataset consisted of estimates of average bite incidence for 83 of 114 districts (Fig 3C), and 58 communes within the catchment of the Moramanga District (Fig 3D, see S2 Text for more details). For the national data, there were two outliers, Toamasina II (the sub-urban district surrounding the city of Toamasina) and Soanierana Ivongo, with higher bite incidence when compared to other districts with similar travel times. While the estimates from the Moramanga data showed higher reported incidence at low travel times at the commune level compared to the district estimates, when aggregated to the district, bite incidence estimates fell within the ranges observed from the national dataset.

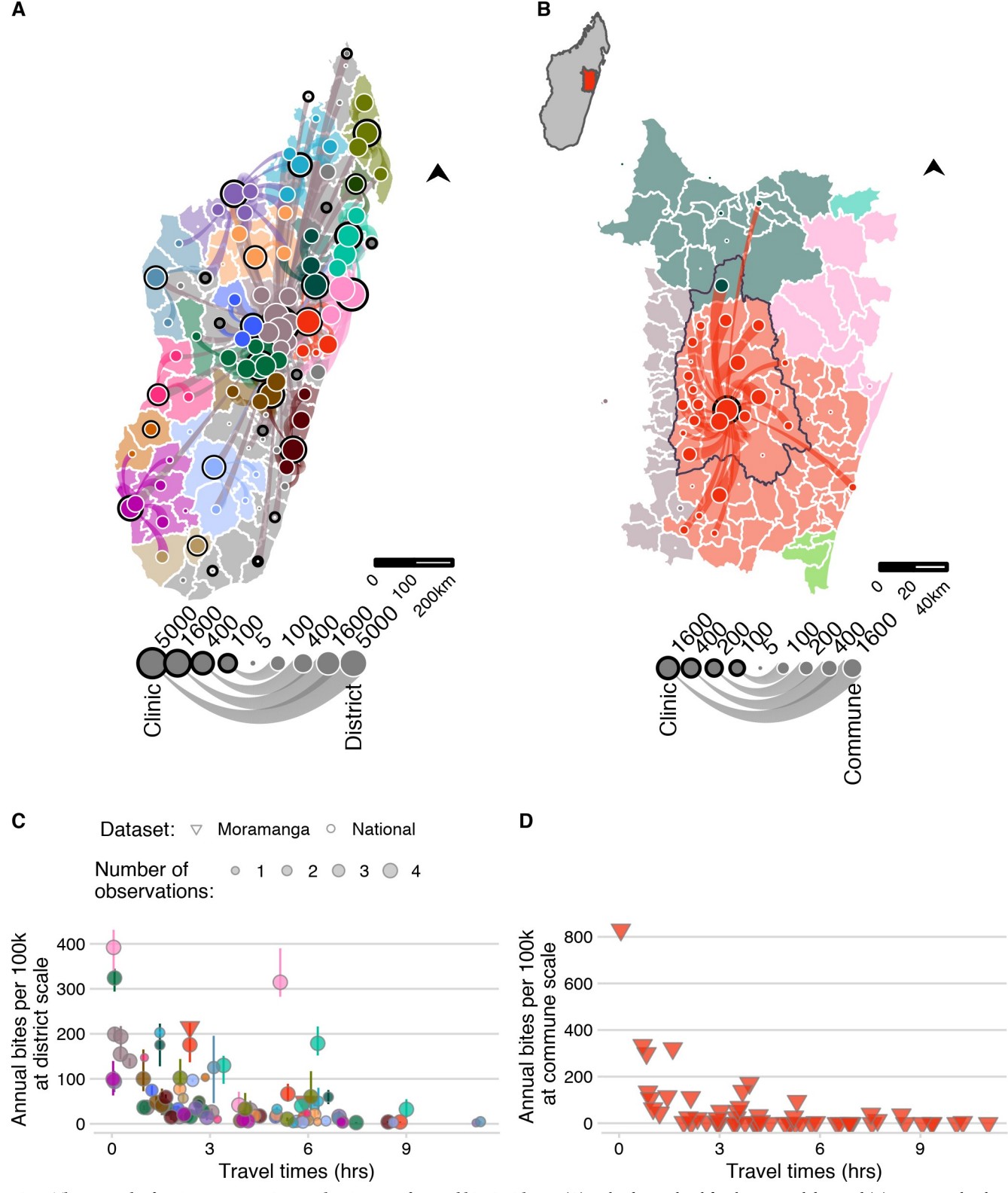

**Fig 3. The network of patient presentations and estimates of annual bite incidence.** (A) at the district level for the national data and (B) commune level for the Moramanga data: circles with a black outline represent the total number of patients reporting to each clinic for which we have data. Color corresponds to the clinic catchment. Circles with a white outline are the total number of bites reported for that administrative unit (plotted as the centroid). Lines show which clinic those patients reported to, with the line width proportional to number of patients from that district reporting to the clinic; flows of

less than 5 patients were excluded. Out-of-catchment reporting is indicated where points and line colors are mismatched. For panel (A) districts in catchments excluded due to lack of forms submitted by the clinic are colored in grey. For (B) the inset of Madagascar shows the location of the enlarged area plotted, which shows the district of Moramanga (outlined in black), all communes included in it's catchment (red polygons), and other communes where bites were reported to colored by their catchment (C) The estimated average annual bite incidence from the national and Moramanga data plotted at the district scale (both datasets) and at the (D) commune scale (Moramanga dataset). Colors correspond to the clinic catchment, shape indicates the dataset, and the size of the point indicates the number of observations (i.e. the number of years for which data was available for the national data; note for Moramanga 33 months of data were used). The point lines indicate the range of estimated bite incidence for each district. Mapped administrative boundaries from OCHA via HDX (https://data.humdata.org/dataset/madagascar-administrative-level-0-4-boundaries, CC-BY-IGO).

Our modeling results show that travel times were a strong and consistent predictor of reported bite incidence in both datasets and across scales with the best fitting models including travel times and an overdispersion parameter (Fig 4, see S3 Text for comparisons to models with catchment effects and with population size as a covariate). As the predictions from the model fit to the Moramanga data without accounting for overdispersion fall within the prediction intervals for the models fit to the national data (Fig 4A), for subsequent predictions, we used the parameter estimates from models fit to the national data, which encompass the range of travel time effects observed in our datasets. Moreover, our out-of-fit predictions to the data across scales suggest that the commune model is able to capture travel time impacts at the commune level (Fig C in S3 Text), therefore we use both the district and commune model to disaggregate burden to the finest scale possible. Finally, we examined the sensitivity of models to how we corrected for underreporting of data and found that parameter estimates of travel time impacts were similar across models and performed similarly in prediction (Figs H and I in S3 Text).

## Current provisioning of PEP substantially reduces human rabies deaths, but incidence of deaths remains high in areas with poor access

In general, the incidence of rabies deaths increases with travel times to clinics, and there is significant sub-national variation when deaths are modeled at the district and commune scale, with the least accessible communities having most deaths (Fig 5B and 5C). We estimate that approximately 800 (95% PI: 600–1000) deaths due to rabies are prevented through PEP each year under the current system of 31 PEP clinics in Madagascar. Overall, we estimate close to 1000 rabies deaths (95% PI: 800–1100) annually in Madagascar. Our estimates vary only slightly depending on the scale of the model (Table 2), but disaggregating deaths to the commune level shows considerable variation in predicted burden within districts (Fig 5A).

## Expanding PEP access to underserved populations is effective at reducing human rabies deaths, but this effect saturates as more clinics provision PEP

We found that targeted expansion of PEP to clinics based on the proportion of the population they reduced travel times for resulted in fewest deaths (Fig A in S5 Text). Here we report results from the commune model, as estimates were consistent across models (Fig 6 and S5 Text). We estimated that strategic PEP expansion to these additional 83 clinics (1 per district) reduced rabies deaths by 19% (95% PI: 14–23%) (Fig 6A). With one clinic per commune (where available, N = 1696), we see a further reduction of 38% (95% PI: 30–46%). However, reductions in burden saturate as more clinics are added (Fig B in S5 Text). Even when all primary clinics provision PEP, our model still predicts 600 (95% PI: 400–800) deaths per annum, and average reporting of rabies exposures remains approximately 66% (95% PI: 33–78%) (Fig E in S5 Text). Our model predicts that as more clinics are added, reported bite incidence saturates (Fig D in S5 Text), and patients shift which clinic they report to (Figs G and H in S5 Text).

Vial demand also outpaces reductions in burden (Fig 6B), with more vials needed per death averted (Fig 6C). Our model predicts an increase from 33,500 vials (95% PI: 22,900–49,400) per

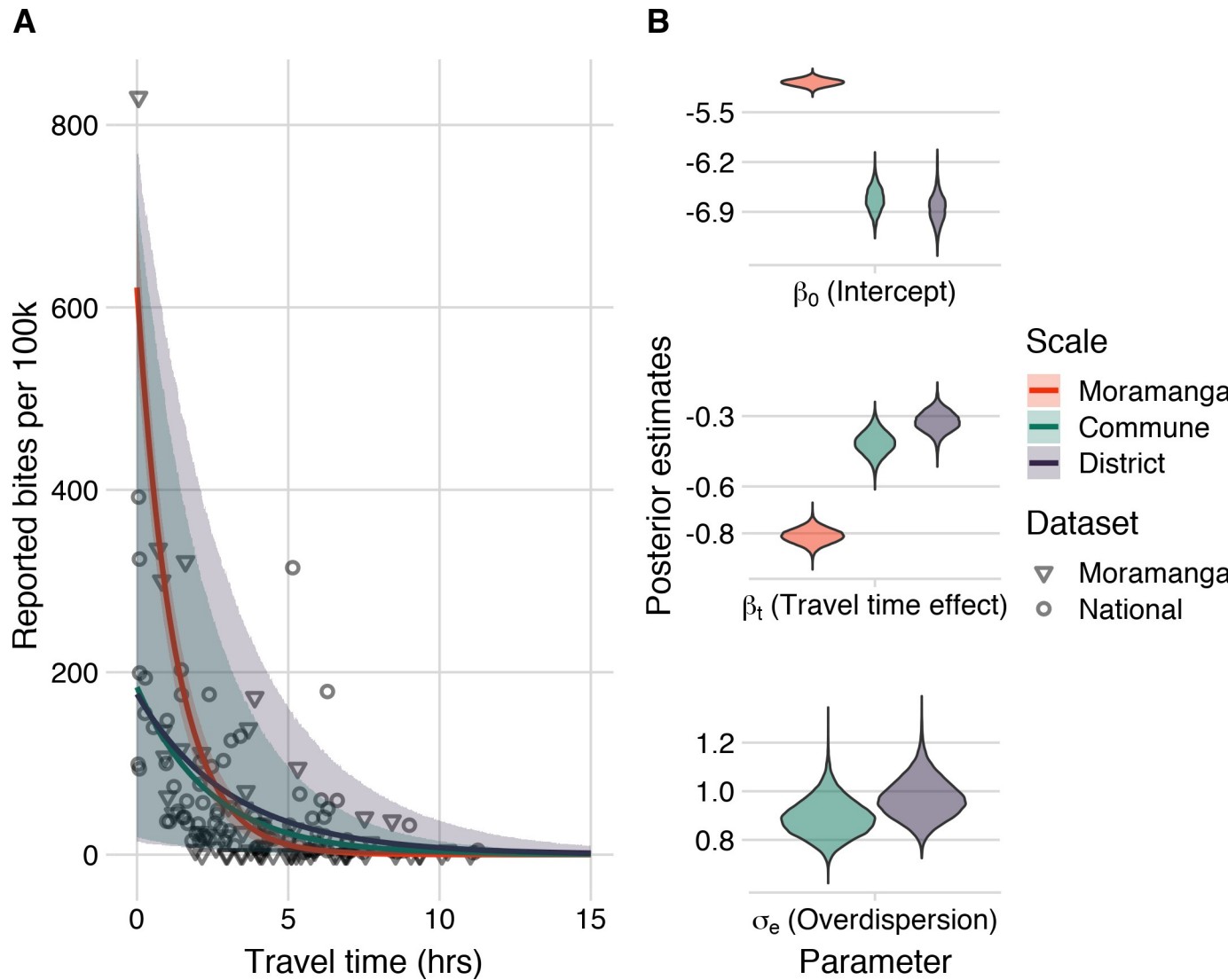

**Fig 4. Travel times as a predictor of reported bite incidence per 100,000 persons.** (A) The estimated relationship between travel time in hours (x-axis) and mean annual reported bite incidence (y-axis). The lines are the mean estimates and the envelopes are the 95% prediction intervals generated by drawing 1000 independent samples from the parameter posterior distributions for three candidate models: model with travel times at the 1) commune- and 2) district-level fitted to the national data with an overdispersion parameter ($\sigma_e$) and 3) travel times at the commune level fitted to the Moramanga data with a fixed intercept and unadjusted for overdispersion. The points show the data: National data (circles) at the district level used to fit the District and Commune models, and Moramanga data (triangles) at the commune level used to fit the Moramanga model. (B) The posterior distribution of parameters from the respective models for the model intercept, travel time effect, and for overdispersion (national data only).

annum under current provisioning but with the abridged intradermal regimen (i.e. visits on days 0, 3, 7), to ~56,900 vials (95% PI: 40200–77800) with 250 clinics providing PEP, and ~86,400 vials (95% PI: 61,600–118,000) if all primary clinics provision PEP. In these scenarios, clinic catchment populations and throughput decrease, with clinics seeing fewer patients per day (Fig F in S5 Text).

## Burden estimates are most sensitive to assumptions of underlying rabies incidence

While qualitative patterns of the current impact of geographic access on human rabies deaths and the impact of expanding access to PEP on reducing these deaths is robust across a wide

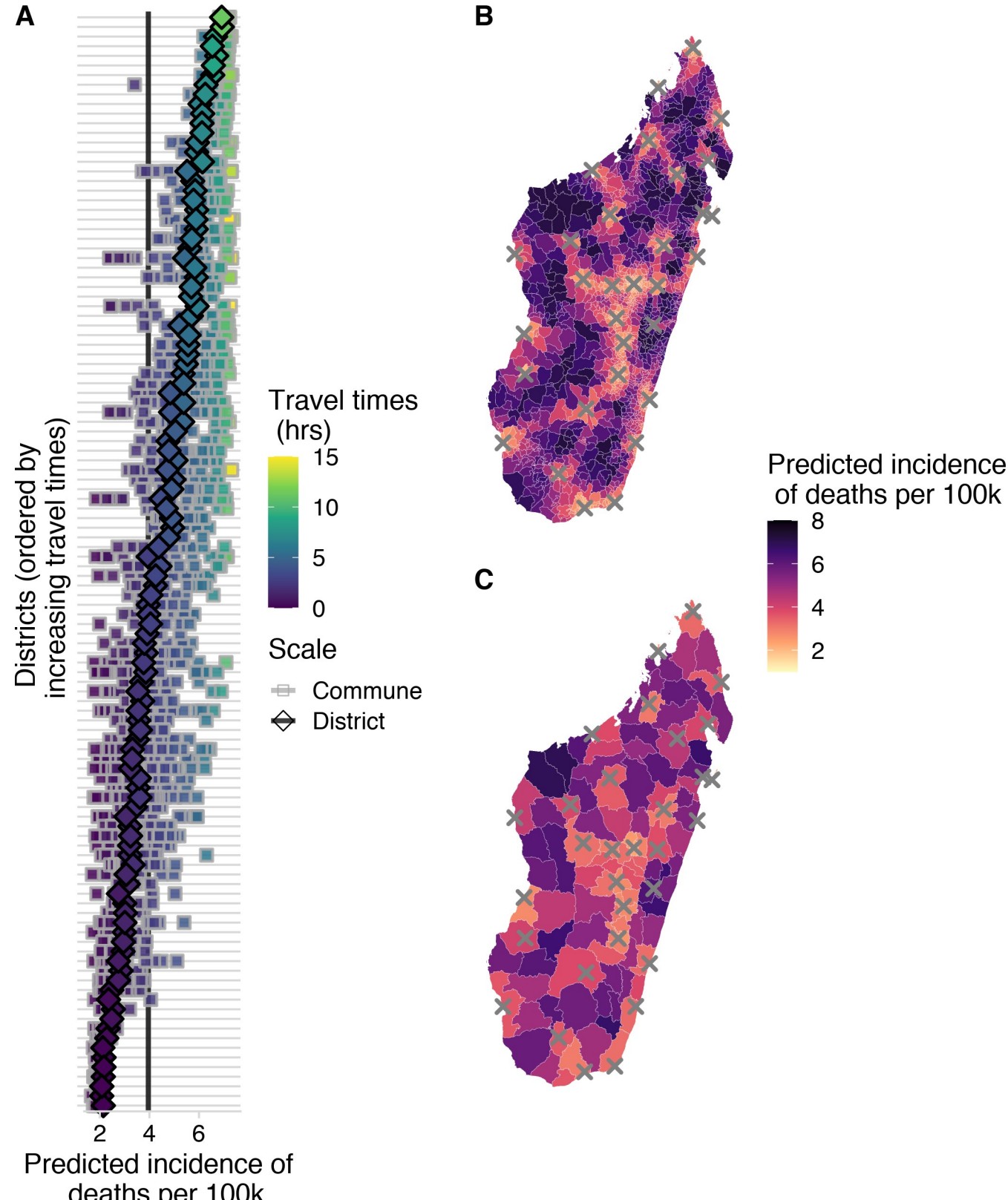

**Fig 5. Spatial variation in predicted incidence of human rabies deaths per 100,000 persons.** (A) for each district (y-axis) in Madagascar. Diamonds show the predicted incidence for the district model and squares show predicted incidence for the commune model fit to the National data for all communes in a

given district. Points are colored and districts ordered by travel times. The vertical lines show the average national incidence of human rabies deaths for the commune (grey) and district (black) models. Incidence mapped to the (B) commune- and (C) district-level from the respective models; grey X's show locations of current clinics provisioning PEP. Mapped administrative boundaries from OCHA via HDX (https://data.humdata.org/dataset/madagascar-administrative-level-0-4-boundaries, CC-BY-IGO).

range of parameter estimates, our sensitivity analyses show that assumptions of the underlying rabies exposure incidence ($E_i$) contribute the most uncertainty to our quantitative estimates (Figs A and B in S6 Text). Uncertainty in bite model parameters contribute less to uncertainty in estimates of burden or impacts of expanded access. For the estimates of vial demand, uncertainty around the model intercept (i.e. the average reported bite incidence) has most impact, rather than the travel time effect or the overdispersion parameter (Fig C in S6 Text). Finally, scaling of incidence with population size (Fig B in S4 Text) modulates the impact of travel times on deaths, with positive scaling of rabies incidence with population size (i.e. more rabies in more populated places) dampening and negative scaling exacerbating the relationship between access and deaths (Fig D in S6 Text). However, the impact of adding clinics remains broadly the same.

## Discussion

### Main findings

We find that the burden of rabies in Madagascar is likely concentrated in areas with poor PEP access. We estimate that current PEP provisioning (at 31 clinics) averts 45% of deaths that would otherwise occur, and that expanding PEP access should reduce mortality, with provisioning in one clinic per district (N = 83), or per commune (N = 1733), expected to reduce mortality by 16% and 33%, respectively. However, improved PEP provisioning is unlikely to eliminate rabies deaths; with over 600 deaths expected even with PEP at all primary clinics (N = 1733). This is partly because travel times remain high (> 2 hours as estimated by the friction surface for over 10% of the population, Fig D in S5 Text) even after expanding PEP to all primary clinics. With reduced travel times, over 20% of exposures will still not seek PEP (Fig E in S5 Text), resulting in ~1.65 rabies deaths per 100,000 people. PEP is expected to remain cost-effective as provisioning expands, to a maximum of 450 USD per death averted (assuming 5 USD per vial), similar to other estimates [4]. While our quantitative predictions depend on assumptions of underlying rabies exposure incidence, the qualitative patterns regarding travel time impacts remain robust and are useful in identifying strategies for provisioning PEP.

### Limitations

Data limitations introduced bias and uncertainty to our estimates. For example, travel times from the Malaria Atlas Project friction surface underestimated patient-reported travel times, with discrepancies between assigned transport speeds (from the Open Street Map user community, or country cluster data [18]) and realities of local travel. In Madagascar, the presence of paved roads does not necessarily reflect their quality or the modes of transport used. Patients seeking PEP at the Moramanga clinic reported various transport methods and highly

**Table 2. Model predictions of average annual reported bite incidence, total deaths, and deaths averted at the national level for the two models (commune level and district level models with travel time predictor and an overdispersion parameter); 95% prediction interval in parentheses.**

| Model | Reported bite incidence per 100k | Burden of deaths | Deaths averted by current PEP provisioning |
|---|---|---|---|
| Commune | 85 (56–129) | 963 (795–1118) | 801 (644–968) |
| District | 85 (52–136) | 958 (752–1156) | 807 (609–1005) |

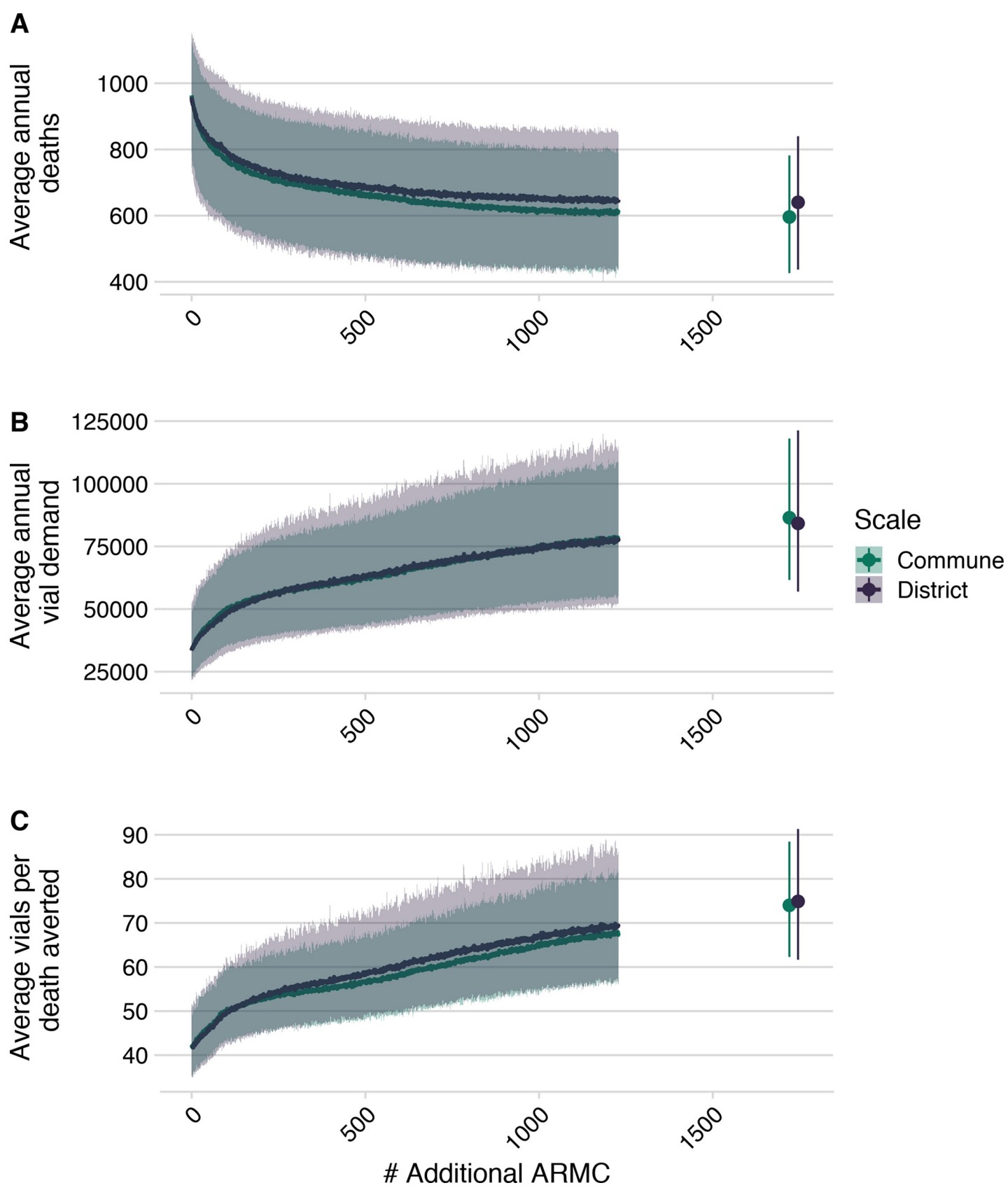

**Fig 6. Impact of expanded PEP access on deaths, deaths averted and vial demand.** (A) Decrease in deaths due to rabies, (B) increase in total number of vials as additional clinics provisioning PEP are added at the national level, and (C) increase in vials needed per death averted based on the two models of reported bite incidence. Lines are the mean of 1000 simulations with envelopes representing 95% prediction intervals. The points show the scenario in which all additional primary clinics and secondary clinics (N = 1733) clinics have been added.

variable travel times even within a single commune. While patient-reported travel times may lack precision from recall and estimation error, they likely better reflect lived experience; validated travel times [30] could improve estimates of spatial health inequities. Similarly, modeled estimates of population distribution [19] also introduce uncertainty. Our analysis of data from the Moramanga District indicates that variation at the sub-district level is high and impacts health seeking behavior. However, we lacked fine-scale data from other catchments for comparison. Additionally, we had to correct for underreporting and incomplete data; strengthening surveillance and routine data collection should improve understanding of health seeking behavior and access, and support monitoring and evaluation of PEP provisioning.

While we rely on a number of assumptions, they are based on data specific to Madagascar or from other similar settings and consistent with estimates from the literature more broadly (see Table 1 and S4 Text). Our burden estimates were most sensitive to assumptions about rabies exposure incidence, drawn from studies in the Moramanga District [15] and elsewhere [4]. As incidence of rabies exposures varies over time and space [31,32], we incorporated uncertainty into our estimates, but we did not find qualitative differences in the effects of travel times on rabies deaths. Our simplifying assumptions regarding patient compliance, which is generally high in Madagascar [15], and on complete efficacy of PEP are unlikely to greatly influence our burden estimates [22]. Likewise, we do not account for differential risk for severely exposed patients not receiving Immunoglobulins (RIG), which is only available at IPM in the capital of Antananarivo, but recent studies show that even in the absence of RIG, PEP is extremely effective [4]. We also assume that clinics reliably provision PEP, but a 2019 KAP survey reported clinics experiencing stock-outs [25].

We assumed geographic access to PEP was the primary driver of health-seeking behavior, but socioeconomic status, education and awareness about rabies [27,33–35] all play a role. For example, most PEP clinics also charge fees (from 0.50–3.00 USD for consultations, wound treatment, etc. [25]) which may also act as barriers to PEP access. In Madagascar, where PEP is free-of-charge, the main cost to patients is transport and time lost. More remote communities tend to be of lower socioeconomic and educational status [2], so travel time may be a proxy for these correlated variables. Significant overdispersion in the data that cannot be explained by travel times suggests that clinic-level variation (e.g. vaccine availability and charges) and regional differences (e.g. dog populations, outbreaks, awareness) further influence health-seeking behavior and vaccine demand. Although our estimates could be improved with better data on rabies incidence, health-seeking behavior, and PEP provisioning, predicting PEP impacts will remain challenging given the complex interactions between socioeconomic factors, access to and quality of care, and human behavior, as illustrated by the case studies in Box 1. However, it is very likely that the impacts of improving access to PEP could be further increased with outreach and awareness raising efforts that we were unable to parameterize.

## Broader context

Recent studies have estimated access to health-seeking behavior and PEP completion and adherence, but not directly linked these estimates to burden [7,36,37]. Our approach for incorporating access to vaccines (echoing [38–42]) into burden estimation methods could guide provisioning of PEP to maximize impacts. This approach will have most value in settings with

## Box 1: Case studies of health seeking behavior for PEP in Madagascar.

1. Anosibe An'ala District (population ~ 100,265), south of Moramanga, has moderate incidence of bite patients (~ 54/100,000 persons) even though travel times often exceed 24 hours. While a road connects the main Anosibe An'ala commune to the Moramanga PEP clinic, it is only passable by large trucks during much of the rainy season, with speeds usually < 10km per hour. Over 9% of patients from Anosibe An'ala had been in close proximity or touched a person that died from rabies (four suspect human rabies deaths of patients who did not receive any PEP), whilst of patients with Category II and III exposures that were interviewed, 11/19 (58%) were bitten by probable rabid dogs. Given the high travel times (although underestimated by the friction surface) and incidence of reported rabies exposures and deaths, we predict a large but unobserved rabies burden in this remote community (~6.02 deaths per year) and we ranked a clinic provisioning PEP in Anosibe An'ala 28th for travel time reductions. Other remote communities likely experience similar high and unrecognized burden, but improved surveillance is necessary to identify such areas. Notably, bite patients in this district demonstrate willingness to travel for free PEP (in some cases walking 3 days to a clinic) with awareness of rabies risk. Community outreach and active surveillance in other remote areas could also greatly improve people's awareness of risk and health seeking behavior.

2. Recently, a middle-aged taxi driver died of rabies in suburban Antananarivo. The day after being bitten by an unknown dog, he reported to a clinic that referred him to the closest clinic provisioning PEP, approximately one hour's drive from his home. His family urged him to go, but he did not believe his risk was high and decided not to seek further care. He developed symptoms two weeks later and was confirmed as a rabies death by the National Rabies Reference Laboratory at IPM. Despite prompt reporting, appropriate referral, and socioeconomic indicators suggesting a high care-seeking probability, this person did not receive PEP. His story highlights the need for sensitization about rabies, how PEP provisioning at peripheral clinics (even in areas with reasonable access) could prevent additional deaths, and ultimately that PEP alone is unlikely to prevent all rabies deaths.

limited PEP access, but will be less valuable where rabies exposures make up a small fraction of patients reporting for PEP e.g. [43,44]. In other settings, similar statistical approaches could be used to identify and quantify key barriers to PEP seeking behavior. For example, reducing the direct cost of PEP is likely to be of more importance than increasing geographic access where PEP costs are high.

Our revised estimate of rabies deaths in Madagascar using this approach was higher than previously estimated (between 280–750 deaths/year) [15], which assumed uniform reporting of 85%, but remained within the range of other empirical and modeling studies from low-income countries [26,27,45–47]. Our estimates of vial demand depend on use of the new abridged intradermal regimen [28], which has been adopted by the Ministry of Health in Madagascar. However, most clinic staff were not aware of WHO classifications of exposure

categories, and vaccination of Category I exposures (those not requiring PEP) remains common practice, comprising 20% of vial demand in Moramanga [15].

We predict that as clinics are expanded, throughput (daily patients reporting to a clinic) will decrease. This may complicate the supply chain and make provisioning PEP more challenging as vial demand becomes less predictable, leading to stock outs or wastage. Decentralized provisioning mechanisms, for example adopting routine childhood vaccine supply chains, or novel vaccine delivery methods such as drones [48], may mitigate these challenges. When nerve tissue vaccines were used in Madagascar, clinics requested vaccines upon demand and PEP access was more widespread, but provisioning the more expensive cell culture vaccines to all clinics became too costly [16]. Widespread vaccine provisioning is therefore feasible given Madagascar's health infrastructure, if cost barriers are removed.

Gavi investment could greatly reduce the access and cost barriers to PEP [6,7,22,49]. Currently, each clinic in Madagascar serves an average catchment of 780,000 persons. Latin American countries, where significant progress has been made towards elimination, aim for one PEP clinic per 100,000 persons. In Madagascar this would require around 212 additional clinics provisioning PEP. We predict that Gavi investment would be highly cost-effective, greatly reducing deaths by expanding PEP supply to underserved areas.

However, our results suggest that PEP expansion alone cannot prevent the majority of rabies deaths, and even given maximal access, achieving 'the last mile,' preventing deaths in the most remote populations, will require disproportionate resources [50]. To achieve 'Zero by 30,' mass dog vaccination will be key to interrupting transmission and eliminating deaths. Integrated Bite Case Management (IBCM) uses bite patient risk assessments to determine rabies exposure status, guide PEP administration, and trigger investigations of rabid animals, potentially identifying other exposed persons [15,51,52]. IBCM is one way to manage PEP effectively [43] and as it relies on exposed persons reporting to clinics, expanding PEP access could strengthen this surveillance framework. These same issues of access, however, apply to both dog vaccination and surveillance, and understanding spatial heterogeneities will be critical to determining how control and prevention interventions can be best implemented [53,54].

## Conclusion

Our study suggests that rabies deaths in Madagascar disproportionately occur in communities with the poorest access to PEP and that expanding PEP access should reduce deaths. Without data on rabies incidence and exposure risk, targeting PEP expansion to underserved areas is a strategic way to reduce rabies burden and provide equitable access, for example, by expanding provisioning to clinics serving populations that target an evidence-based travel time threshold or catchment size. Implementing outreach programs to raise awareness should further increase the efficacy of PEP expansion by improving care seeking. Better surveillance is also needed to understand the geographical distribution of rabies exposures and identify populations most at risk, and to evaluate the effectiveness of PEP expansion at preventing human rabies deaths. Gavi investment could support countries to more equitably provision PEP and overcome barriers to access ([9], see Box 1 for case studies), but as PEP alone cannot prevent all rabies deaths, investment should be used to catalyze mass dog vaccination to interrupt transmission, and eventually eliminate rabies deaths.

## Supporting information

**S1 Text. Estimating travel times to the closest clinic provisioning PEP. Fig A. Raster inputs to estimate travel times to the closest clinic provisioning PEP for Madagascar.** (A) Friction surface of travel speeds (in minutes per meter) at an ~ 1x1 km scale, with location of current

clinics provisioning PEP (N = 31) shown with black x's (original source: Malaria Atlas Project friction surface, (https://malariaatlas.org/research-project/accessibility-to-cities/, CC-BY 3.0). (B) Population estimates resampled to the same friction surface (original source: WorldPop, https://www.worldpop.org/geodata/summary?id=70, CC-BY 4.0 license). **Fig B. Raw patient reported and driving travel time data.** (A) Distribution of travel times estimated at the grid cell level and reported by patients for each commune where patient data were available from the Moramanga PEP clinic (B) Reported driving times between locations, with the color corresponding to the total driving time and the size of the line showing the direction of travel (narrow to wide ~ origin to destination). Paths are Bezier curves from origin to destination, and do not show actual paths driven. Administrative boundaries from OCHA via HDX (https://data.humdata.org/dataset/madagascar-administrative-level-0-4-boundaries, CC-BY-IGO). **Fig C. Reported modes of transport used compared to reported travel times for patients reporting to the Moramanga PEP clinic.** Fig D. Observed estimates of travel times (commune means of patient reported travel times and driving times between point locations) vs. estimates from friction surface. Predicted by (A) Distance (km) (Euclidean distance between origin and destination for driving times and distance from the commune centroid to the Moramanga PEP clinic for commune means) (B) Travel time estimates and (C) Travel time estimates weighted by population (for commune means only). Administrative boundaries are from OCHA (https://data.humdata.org/dataset/madagascar-administrative-level-0-4-boundaries, CC-BY-IGO). **Table A. $R^2$ values from linear models with estimated access metrics predicting either driving times or commune means of patient reported travel times. Fig E. Estimates of mean travel times weighted by population at the (A) District (B) Commune scale. Administrative boundaries from OCHA via HDX** (https://data.humdata.org/dataset/madagascar-administrative-level-0-4-boundaries, CC-BY-IGO).
(PDF)

**S2 Text. Estimating bite incidence. Fig A. Catchment assignments by travel time.** (A) Catchments as assigned by closest clinic for the majority of the population within a commune (polygon fill) or within a district (polygon outline). Admin units where the fill and border colors do not match show places where assigned catchments differ at the district vs commune scale. (B) Distribution of the proportion of the population in a given administrative unit (district or commune) served by the catchment assigned. (C) The proportion of bites reported to each clinic which originated from a district within the assigned catchment. The vertical line indicates the proportion of bites from within the assigned catchment for the Moramanga data (~90%). Administrative boundaries from OCHA via HDX (https://data.humdata.org/dataset/madagascar-administrative-level-0-4-boundaries, CC-BY-IGO). **Fig B. Estimating undersubmission of patient forms.** (A) Number of patient forms submitted to IPM for each clinic over the study period for each clinic, with periods of time where no forms were submitted for > = 15 days excluded (in grey); (B) Estimates for the proportion of forms submitted for each clinic (points are the estimate for each year and the line is the range), calculated as the # of days in a year which were not excluded based on the criteria of 15 consecutive days of nonsubmission/365. (C) The difference between log(estimated) and log(observed) vials provisioned for the period of 2013–2017 for each clinic correcting for under-submission (squares) using the 15 day cut-off show in A and B, vs. not correcting for under-submission (circles). We did not have data on vials provisioned for IPM. **Table A. Root mean squared error (MSE) between observed vials provisioned and estimated by the different consecutive day threshold for correcting for periods of no form submission, with the minimum root MSE in bold.**
(PDF)

**S3 Text. Modeling reported bite incidence. Fig A. Correlation between travel time in hours (the average weighted by the population) and population size of administrative units at the district and commune scale. Table A. DIC and convergence estimates (maximum potential scale reduction factor and multivariate psrf, values < 1.1 indicate convergence) for all models.** For the column pop effect, addPop = models with population size as additional covariate, onlyPop = models with population as only covariate, flatPop = models with population as offset in model. For the intercept type: random = random intercept by catchment, fixed = a single fixed intercept was estimated). The Overdispersion column indicates whether an overdispersion parameter was estimated (yes) or not (no). **Fig B. Prediction to data used to fit each model.** Log of the observed bites against the log of predicted bites generated from sampling 1000 independent draws from the posterior distributions for each parameter, with the points the mean of the predictions and the linerange the 95% prediction intervals. Columns are by the type of model intercept (either a fixed intercept or a random intercept by catchment) and rows are the type of model structure with respect to the population covariate (addPop = population size as additional covariate, onlyPop = population as only covariate, flatPop = population as offset in model). Colors show which data set was used for fitting and the scale of the model (Moramanga = Moramanga data with covariates at the commune level, Commune = National data with covariates at the commune level, District = National data with covariates at the district level). **Fig C. Out of fit predictions to data.** Log of the observed bites against the log of predicted bites for data not used to fit the model. Predictions were generated by sampling 1000 independent draws from the posterior distributions for each parameter, with the points the mean of the predictions and the linerange the 95% prediction intervals. The first two columns are the predictions from the commune and district model fitted to the national data for the Moramanga data with fixed and random intercepts. The third column are predictions from models fitted to the Moramanga data for the national data at the commune and district scale (only fixed intercept models). Rows are the type of model structure with respect to the population covariate and colors show which data set was used for fitting as per Fig B in S3 Text. **Fig D. Posterior estimates of parameters from models with travel time and population as an offset.** Comparing models accounting for overdispersion ($\sigma_e$) compared to models with no overdispersion parameter (flatPop in Figs B and C in S3 Text). For the Moramanga model, as data came from a single catchment, models with a random catchment effect ($\sigma_0$) were not fitted. **Fig E. Predicted relationship between travel times (in hours) and reported bite incidence per 100,000 persons.** Generated from sampling 1000 independent draws from the posterior distributions for each parameter, with the line the mean of the predictions and the envelopes showing the 95% prediction intervals. Rows are by the type of model intercept (either a fixed intercept or a random intercept by catchment) and columns are whether the model estimated an overdispersion parameter. The points show the data used to fit the models (the National dataset), as well as the Moramanga dataset. Note the different y-axis limits between the fixed and random intercept models. Fig F. Posterior estimates of the catchment intercepts ($\alpha$ parameters, with $\beta_0$ the estimated mean intercept) for models with and without an overdispersion parameter. **Fig G. Predicted relationship between travel times (in hours) and reported bite incidence per 100,000 persons.** For random intercept model without overdispersion vs. fixed intercept model with overdispersion, generated from sampling 1000 independent draws from the posterior distributions for each parameter, with the line the mean of the predictions and the envelopes showing the 95% prediction intervals. The points show the data used to fit the models (the National dataset), as well as the Moramanga dataset. **Fig H. Posterior estimates for models with population as an offset and an overdispersion parameter.** Columns show estimates from models fitted to the national dataset (1) corrected for both under-submission by correcting for periods of at least 7 days where zero

patient forms were submitted, (2) correcting for periods of 15 days where zero patient forms were submitted, and (3) with the raw data not correcting for under-submission. **Fig I. The estimated relationship between travel time in hours (x-axis) and mean annual bite incidence per 100,000 persons (y-axis).** For models with population as an offset and an overdispersion parameter. Panels show predictions from models fitted to the national dataset 1) correcting for periods of at least 7 days where zero patient forms were submitted (a less stringent cutoff resulting in lower estimates of the proportion of forms submitted and thus higher estimates of reported bite incidence), (2) correcting for periods of 15 days where zero patient forms where submitted, as presented in the main analysis, and (3) with the raw data not correcting for under-submission (resulting in lower estimates of reported bite incidence). Predictions were generated from sampling 1000 independent draws from the posterior distributions for each parameter, with the line the mean of the predictions and the envelopes showing the 95% prediction intervals. The points show the data used to fit the models.
(PDF)

**S4 Text. Range of rabies exposure incidence in people. Fig A. Estimated exposures per 100,000 given a range of human to dog ratios (HDRs, x-axis) and annual dog rabies incidence (y axis).** Assuming that each dog on average exposes 0.39 persons. The black dashed lines show the range of human to dog ratios (HDRs) we use in the main analysis to estimate the range of human exposure incidence (where the red horizontal line and black dashed lines intersect). The grey dashed lines show the HDRs estimated from the Moramanga district from a recent study. The cells with red outlines show the range of estimated exposure incidence from a previous study of bite patients in Moramanga District. Administrative boundaries from OCHA via HDX (https://data.humdata.org/dataset/madagascar-administrative-level-0-4-boundaries, CC-BY-IGO). **Fig B. Range of constrained scaling factors generated for district and commune population size.** Underlying rabies exposures either (A) decreases with increasing population size or (B) increases with increasing population size across a fixed range of exposure incidence (15.6–76 exposures/100k persons). Lines show the expected relationship, with points showing where administrative units fall along this curve, and maps show how this results in variation in assumed exposure incidence spatially at the commune and district level.
(PDF)

**S5 Text. Estimating the impact of expanding PEP provisioning to additional clinics. Fig A. Comparing metrics for ranking clinics for targeted expansion.** We simulated expansion using three different ranking metrics: 1) reduction in mean travel times (green line) 2) the proportion of the population for which travel times were reduced (red dashed line) and 3) the proportion of the population for which travel times were reduced weighted by the reduction in travel times (pink dashed line). For each of these, we simulated burden using our decision tree framework (y axis is the mean of 1000 simulations of annual deaths at the national level). The blue lines show 10 simulations of randomly expanding access on reducing burden as a comparator. The panels show to the commune and district model of reported bite incidence. **Fig B. Map of the location and at what step each clinic was added.** The circles are each of the primary clinics across the country sized by the resulting average reduction in burden (based on smoothed annual burden estimates from the commune model, see inset). The large white crosses show the location of the existing 31 clinics provisioning PEP in Madagascar and the smaller white crosses are the additional primary clinics in the country which were added in the final step but not ranked. Administrative boundaries from OCHA via HDX (https://data.humdata.org/dataset/madagascar-administrative-level-0-4-boundaries, CC-BY-IGO). **Fig C. Maps of how travel times change as clinics are added.** (A) at the ~1 x 1 km grid cell (B)

commune and (C) district scales. The columns are ordered by the number or clinics at each step: baseline (N = 31), + 83 (1 per district), + 200, + 600, + 1406 (1 per commune), and max (+ 1696 clinics, all additional primary clinics in the country). Grey pixels show the location of clinics provisioning PEP at each step. Commune and district values are the average grid cell travel times weighted by the population in each cell. Administrative boundaries from OCHA via HDX (https://data.humdata.org/dataset/madagascar-administrative-level-0-4-boundaries, CC-BY-IGO). **Fig D. Shifts in key metrics as clinics are added.** (A) travel times (hrs, x-axis is square root transformed), (B) bite incidence per 100,000 persons and boxplots showing the median for communes and district models (colors). **Fig E. Shifts in key metrics as clinics are added.** (A) reporting, (B) death incidence per 100,000 persons for the commune and district models (colors) as clinics are added. **Fig F. Shifts in key metrics as clinics are added.** (A) catchment population size, (B) annual vial demand, and (C) daily throughput (i.e. average number of patients reporting each day) given estimates of bite incidence for the commune and district models (colors). For vial demand estimation, catchment population sizes are the same for each model as these populations are allocated at the grid cell level (i.e. population in a grid cell is allocated to the clinic catchment it is closest to in terms of travel times regardless of district or commune). All x-axes are log transformed. **Fig G. Shifts in where bites are reported to as clinics are added for the commune model.** The circles show the clinic locations for each scenario, with size proportional to the annual average bites reported to that clinic. Lines show where the bites are reported from (commune centroid) also proportional to the number of bites. The polygon shading shows the commune level reported bite incidence. Administrative boundaries from OCHA via HDX (https://data.humdata.org/dataset/madagascar-administrative-level-0-4-boundaries, CC-BY-IGO). **Fig H. Shifts in where bites are reported to as clinics are added for the district model.** The circles show the clinic locations for each scenario, with size proportional to the annual average bites reported to that clinic. Lines show where the bites are reported from (commune centroid) also proportional to the number of bites. The polygon shading shows the district level reported bite incidence. Administrative boundaries from OCHA via HDX (https://data.humdata.org/dataset/madagascar-administrative-level-0-4-boundaries, CC-BY-IGO). (PDF)

**S6 Text. Sensitivity of burden estimates to parameter assumptions. Fig A. Sensitivity analyses for baseline burden estimates.** (A) Range of parameters used in the univariate sensitivity analysis (upper 97.5% and lower 2.5% credible interval of posterior for the parameters in the bite incidence model, 95% CI of estimate from [1] for $\rho_{max}$, and fixed at the minimum and maximum of the range used in the main analysis for all other parameters) (B) Range of estimates of the estimate average deaths at the national level for range of parameter estimates (point shows the estimate presented in the main analyses and the ends show the upper and lower estimates for the parameter range) and (C) estimates for the predicted relationship between travel times and incidence of human rabies deaths across the same parameter ranges (line shows the estimate presented in the main analyses and the envelope shows the upper and lower estimates for the parameter range) for the two models (colors). **Fig B. Sensitivity analysis for PEP expansion.** (A) Estimates of the maximum proportion reduction of deaths from the baseline for range of parameter estimates (point shows the estimate presented in the main analyses and the ends show the upper and lower estimates for the parameter range) and (B) estimates for how human rabies decreases proportional to the baseline as clinics are added for these same parameter ranges (line shows the mean estimate presented in the main analyses and the envelope shows the mean estimates for the parameters fixed at the upper and lower range) for the two models (colors). See Fig A for ranges used for each parameter. **Fig C.**

**Sensitivity analysis for vial demand.** (A) Estimates of the baseline vial demand for where each parameter was set to the upper and lower estimate of parameter estimates (point shows the estimate presented in the main analyses and the ends show the upper and lower estimates for the parameter range) and (B) estimates for how vial demand increases as clinics are added for these same parameter ranges (line shows the mean estimate presented in the main analyses and the envelope shows the mean estimates for the parameters fixed at the upper and lower range) for the two models (colors) for the two models (colors). See Fig A for ranges used for each parameter. **Fig D. Sensitivity analysis for exposure scaling with population.** Predicted relationship between (A) travel times and rabies death incidence per 100k persons and (B) the proportional reduction in deaths from the baseline as clinics are added for the district model vs. commune models (colors) with columns comparing the baseline (presented in main analyses) and assumptions of rabies exposure incidence increasing or decreasing with human population size (shapes). The points show the mean estimates from 1000 simulations and the line ranges show the 95% prediction intervals.
(PDF)

**S7 Text. Details on software packages used.**
(PDF)

## Acknowledgments

We thank all the clinicians and staff at the clinics across the country. We are grateful to IPM and the Ministry of Public Health who collect and maintain data on PEP provisioning. In particular, we thank the GIS unit for assistance with spatial data, Michael Luciano Tantely for sharing driving time data, and Claire Leblanc, Rila Ratovoson, and Daouda Kassie for sharing results of their work in the Moramanga District. In addition, we thank Jean Hyacinthe Randrianarisoa, Ranaivoarimanana, Fierenantsoa Randriamahatana, Esther Noiarisaona, Cara Brook, Amy Winter, Christian Ranaivoson, John Friar, and Amy Wesolowski for assistance.

## Author Contributions

**Conceptualization:** Malavika Rajeev, Glenn Torrencelli Edosoa, Fleur Hierink, Laurence Baril, C. Jessica E. Metcalf, Katie Hampson.

**Data curation:** Malavika Rajeev, Hélène Guis, Glenn Torrencelli Edosoa, Chantal Hanitriniaina, Anjasoa Randrianarijaona, Reziky Tiandraza Mangahasimbola, Ravo Ramiandrasoa, José Nely, Jean-Michel Heraud, Soa Fy Andriamandimby, Laurence Baril.

**Formal analysis:** Malavika Rajeev, Anjasoa Randrianarijaona, Reziky Tiandraza Mangahasimbola, Fleur Hierink, Katie Hampson.

**Funding acquisition:** Malavika Rajeev, C. Jessica E. Metcalf, Katie Hampson.

**Investigation:** Malavika Rajeev, Glenn Torrencelli Edosoa, Chantal Hanitriniaina, Soa Fy Andriamandimby, Laurence Baril.

**Methodology:** Malavika Rajeev, Hélène Guis, Chantal Hanitriniaina, Anjasoa Randrianarijaona, Reziky Tiandraza Mangahasimbola, Fleur Hierink, Laurence Baril, C. Jessica E. Metcalf, Katie Hampson.

**Project administration:** Malavika Rajeev, Hélène Guis, Glenn Torrencelli Edosoa, Chantal Hanitriniaina, Anjasoa Randrianarijaona, Reziky Tiandraza Mangahasimbola, Ravo Ramiandrasoa, José Nely, Jean-Michel Heraud, Soa Fy Andriamandimby, Laurence Baril.

**Resources:** Malavika Rajeev, Hélène Guis, Glenn Torrencelli Edosoa, José Nely, Jean-Michel Heraud, Soa Fy Andriamandimby, Laurence Baril, C. Jessica E. Metcalf, Katie Hampson.

**Software:** Malavika Rajeev, Anjasoa Randrianarijaona, Reziky Tiandraza Mangahasimbola.

**Supervision:** Hélène Guis, Glenn Torrencelli Edosoa, Jean-Michel Heraud, Soa Fy Andriamandimby, Laurence Baril, C. Jessica E. Metcalf, Katie Hampson.

**Validation:** Malavika Rajeev.

**Visualization:** Malavika Rajeev.

**Writing – original draft:** Malavika Rajeev.

**Writing – review & editing:** Malavika Rajeev, Hélène Guis, Glenn Torrencelli Edosoa, Chantal Hanitriniaina, Anjasoa Randrianarijaona, Reziky Tiandraza Mangahasimbola, Fleur Hierink, Ravo Ramiandrasoa, José Nely, Jean-Michel Heraud, Soa Fy Andriamandimby, Laurence Baril, C. Jessica E. Metcalf, Katie Hampson.

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
