## [Decision Letter · Decision Letter 0]

20 Dec 2020

Dear Ms Rajeev,

Thank you very much for submitting your manuscript "How geographic access to care shapes disease burden: the current impact of post-exposure prophylaxis and potential for expanded access to prevent human rabies deaths in Madagascar" for consideration at PLOS Neglected Tropical Diseases. As with all papers reviewed by the journal, your manuscript was reviewed by members of the editorial board and by several independent reviewers. The reviewers appreciated the attention to an important topic. Based on the reviews, we are likely to accept this manuscript for publication, providing that you modify the manuscript according to the review recommendations. 

Sincerely,

Simon Rayner

Associate Editor

Robert Reiner

Deputy Editor

Reviewer's Responses to Questions

**Key Review Criteria Required for Acceptance?**

**Methods**

-Are the objectives of the study clearly articulated with a clear testable hypothesis stated?

-Is the study design appropriate to address the stated objectives?

-Is the population clearly described and appropriate for the hypothesis being tested?

-Is the sample size sufficient to ensure adequate power to address the hypothesis being tested?

-Were correct statistical analysis used to support conclusions?

-Are there concerns about ethical or regulatory requirements being met?

Reviewer #1: The authors performed the well-designed epidemiological analysis; given the distance from the residence to the PEP accessibility and assess the rabies risk. There is only once concern arising how much the prevalence (or exposure to residence) of rabid dogs in each area was included into this study. The authors also pointed out the uncertainty of underlying rabies expose according to the sensitivity analysis. This should be more clearly describe in the manuscript, with a validity of the corresponding parameters used in the current study.

Regarding Lines 275-281, the authors indicate the bases with adequate references how to reach the idea that human-to-dog rabies ratio are both positively and negatively correlated with human populations.

Reviewer #2: (No Response)

Reviewer #3: All the criterias are met. Minor points are the following:

- Please indicate in S1.1A that the black crosses represent ARMCs.

- Delete in line 132 'the'

- Fig. 1 with its description is confusing. I would rather recommend a list with the variables and their description.

- Line 234, can this really be assumed? A baseline study in West Africa demonstrated, that only half of the bite victims complete PEP. However, based on the information regarding the high patient compliance in Madagaskar (line 539), I think you can make this assumption here.

**Results**

-Does the analysis presented match the analysis plan?

-Are the results clearly and completely presented?

-Are the figures (Tables, Images) of sufficient quality for clarity?

Reviewer #1: There are no points to be issued on Results part.

Reviewer #2: The most important result is "Estimating the impact of PEP provisioning." I think it would be good to have these results summarized in a table. This table could include/replace table 2.

Reviewer #3: The analysis presented match the analysis plan and the results are clearly and completely presented. The figures have sufficient quality.

**Conclusions**

-Are the conclusions supported by the data presented?

-Are the limitations of analysis clearly described?

-Do the authors discuss how these data can be helpful to advance our understanding of the topic under study?

-Is public health relevance addressed?

Reviewer #1: Overall, the statements in Conclusion part is adequate and convince readers.

Reviewer #2: Out-of-stock is briefly mentioned in this section but I think it is worth to discuss also that the supply chain is getting much more complicated if PEP hast o be delivered to much more clinics.

Reviewer #3: Discussion:

-Remove a % in line 493

This work nicely demonstrates that even with provisioning PEP all over the country, there will still be rabies deaths in the country. So besides the provision of PEP for humans, the disease needs to be eliminated in the animal reservoir.

**Editorial and Data Presentation Modifications?**

Reviewer #1: No

Reviewer #2: (No Response)

Reviewer #3: -

**Summary and General Comments**

Reviewer #1: In Lines 523-525, the authors discuss the potential risk factor of human rabies in Madagascar. These are very general and commonly argued in anywhere. Don’t the authors can find the supportive reasons or data for these assumptions? Specifically, box1 may provide the example of association between area and awareness to the risk. We would like to know how these gaps may arise.

Reviewer #2: The authors describe an approach to model the efficacy of different PEP distribution scenarios scenarios interesting and new. The approach is based on several steps and each steps is complex and relies on a set of assumptions. Because it would be difficult to collect empirical data (e.g. in a cluster randomized trial), I think the study is worth publishing. I appreciate that a lot of details and supplementary information is provided and that the source code and data is public. My main critic points are: i) due to the complexity and amount and diversity of information (the appendix alone has 47 pages) the authors should guide the reader better through the manuscript. Some suggestions are provided below, ii) the results are based on many assumptions – I am not sure if the authors are not a bit too optimistic with their approach. I think the next step should be to try to validate the model before "Our framework could be used to guide PEP expansion and improve targeting of interventions."

Ideas to guide the reader a bit better through the manuscript:

- One weakness oft he paper is that the reader has to jump back and forth to get all the information. Different parameters are introduced over 5 pages. Maybe the method section could be rearranged and a bit sharpened. E.g. p[rabid] is introduced in line 201. In line 208 (a figure legend) it is explained that R[rabid] is constrained. In line 225 it is explained which range was used to express uncertainty about P[rabies], in line 229 the constraining formula is provided and finally in Table 1 the values distribution and reference for this parameter is provided. I don't know if the constraint is so important that it has to be mentioned to this extend. ;Maybe a footnote in table1 would be sufficient. If not, there is certainly a better way to arrange the method section that it can be introduced at only one (maybe 2) occasion. 

- Same with bite incidence. The result section shows some graphs and a lot of information related to model fit. But the reader is likely more interested in the estimated mean incidence. But this value is reported is not in this but in the next paragraph.

- Terminology could be simplified. E.g. using consistently either commune or CSB2 in results and discussion to describe this level 

- The appendix is quite long. A table of content would be helpful

- Isn't it possible to integrate the information of tables S6.1/2 into figures S6.1/2. 

Abstract

- After reading the abstract I had no clear impression what the authors exatly did, e.g. ehat kind of data they used to estimate travel time, bite incidence and rabies deaths.

-"expanding PEP to one clinic per district could reduce deaths by 19%" Please add the number of districts (114) or the number of additional clinics (83). 

Authors summary

" but our results suggest that expansion alone will not eliminate deaths." Well, how could PEP eliminate human rabies deaths? It does not interrupt transmission. I think this statement should be rephrased.

Travel time: The authors used geographical information to estimate theoretical travel time as an proxi for access. In addition they tried to validate the approach by comparing the estimates with travel time reported by patients and by travel time reported by the Institute Pasteur. However, very basic information is lacking, e.g. I could even not find the number of patients interviewed and it remains unclear what " driving times collected by IPM during field missions " actually means. Of course, reported and estimated travel times are correlated – both are associated with distance – but if I look at figure 2c or the appendix the variation is remarkable. Some estimate of fit would be nice. Maybe mean/median residual or something similar. By the way: I don't think the shaded areas (CIs around the regression lines) does not add a lot tot he interpretation. I would remove them because the points below are difficult to see. 

Figure2.2: A indicates " (A) The daily time series of the number of forms submitted by each clinic," I don't think that the figure presents time series data. B boxplots for 4 points are (literally?) a bit pointless. Present only the 4 points or the 4 points + mean or median.

Minor points 

Is the overdisperion parameter presented in Fig 4 the estimate or log(estimate)? If it is the estimate, an estimate below 1 is rather unusual. 

Lines 131 133: please use superscript instead of caret to indicate exponentiation

Lines 493: 45%%

Lines 494-494: Please keep the same order.

Reviewer #3: It was a pleasure to read this fantastic work.

PLOS authors have the option to publish the peer review history of their article (what does this mean?). If published, this will include your full peer review and any attached files.

Reviewer #1: No

Reviewer #2: Yes: Jan Hattendorf

Reviewer #3: No
---

## [Decision Letter · Decision Letter 1]

24 Feb 2021

Dear Ms Rajeev,

We are pleased to inform you that your manuscript 'How geographic access to care shapes disease burden: the current impact of post-exposure prophylaxis and potential for expanded access to prevent human rabies deaths in Madagascar' has been provisionally accepted for publication in PLOS Neglected Tropical Diseases.

Best regards,

Simon Rayner

Associate Editor

Robert Reiner

Deputy Editor

Reviewer's Responses to Questions

**Key Review Criteria Required for Acceptance?**

**Methods**

-Are the objectives of the study clearly articulated with a clear testable hypothesis stated?

-Is the study design appropriate to address the stated objectives?

-Is the population clearly described and appropriate for the hypothesis being tested?

-Is the sample size sufficient to ensure adequate power to address the hypothesis being tested?

-Were correct statistical analysis used to support conclusions?

-Are there concerns about ethical or regulatory requirements being met?

Reviewer #1: (No Response)

**Results**

-Does the analysis presented match the analysis plan?

-Are the results clearly and completely presented?

-Are the figures (Tables, Images) of sufficient quality for clarity?

Reviewer #1: (No Response)

**Conclusions**

-Are the conclusions supported by the data presented?

-Are the limitations of analysis clearly described?

-Do the authors discuss how these data can be helpful to advance our understanding of the topic under study?

-Is public health relevance addressed?

Reviewer #1: (No Response)

**Editorial and Data Presentation Modifications?**

Reviewer #1: (No Response)

**Summary and General Comments**

Reviewer #1: (No Response)

PLOS authors have the option to publish the peer review history of their article (what does this mean?). If published, this will include your full peer review and any attached files.

Reviewer #1: No

---

## [Editor Report · Acceptance letter]

20 Apr 2021

Dear Ms Rajeev,

We are delighted to inform you that your manuscript, "How geographic access to care shapes disease burden: the current impact of post-exposure prophylaxis and potential for expanded access to prevent human rabies deaths in Madagascar," has been formally accepted for publication in PLOS Neglected Tropical Diseases.

Best regards,

Shaden Kamhawi

co-Editor-in-Chief

Paul Brindley

co-Editor-in-Chief
